# Kinetics of the Thermal Degradation of Poly(lactic acid) and Polyamide Bioblends

**DOI:** 10.3390/polym13223996

**Published:** 2021-11-19

**Authors:** Félix Carrasco, Orlando Santana Pérez, Maria Lluïsa Maspoch

**Affiliations:** 1Department of Chemical Engineering, Universitat de Girona (UdG), C/Maria Aurèlia Capmany 61, 17003 Girona, Spain; 2Centre Català del Plàstic (CCP), Universitat Politècnica de Catalunya Barcelona Tech (UPC-EEBE), C/Colom 114, 08222 Terrassa, Spain; orlando.santana@upc.edu (O.S.P.); maria.lluisa.maspoch@upc.edu (M.L.M.)

**Keywords:** PLA, PA, bioblend, thermal stability, kinetic models, reaction mechanisms, random scission

## Abstract

Poly(lactic acid) (PLA) and biosourced polyamide (PA) bioblends, with a variable PA weight content of 10–50%, were prepared by melt blending in order to overcome the high brittleness of PLA. During processing, the properties of the melt were stabilized and enhanced by the addition of a styrene-acrylic multi-functional-epoxide oligomeric reactive agent (SAmfE). The general analytical equation (GAE) was used to evaluate the kinetic parameters of the thermal degradation of PLA within bioblends. Various empirical and theoretical solid-state mechanisms were tested to find the best kinetic model. In order to study the effect of PA on the PLA matrix, only the first stage of the thermal degradation was taken into consideration in the kinetic analysis (*α* < 0.4). On the other hand, standardized conversion functions were evaluated. Given that it is not easy to visualize the best accordance between experimental and theoretical values of standardized conversion functions, an index, based on the integral mean error, was evaluated to quantitatively support our findings relative to the best reaction mechanism. It was demonstrated that the most probable mechanism for the thermal degradation of PLA is the random scission of macromolecular chains. Moreover, *y*(*α*) master plots, which are independent of activation energy values, were used to confirm that the selected reaction mechanism was the most adequate. Activation energy values were calculated as a function of PA content. Moreover, the onset thermal stability of PLA was also determined.

## 1. Introduction

The generation of polymers derived from renewable sources, also called bio-based polymers, is an important field of research due to the role that these ecofriendly polymers play in reducing plastic residues, which are a source of pollution, and carbon dioxide production, which leads to a decrease in the carbon footprint of its lifecycle [1,2]. Over the past decade, bio-based polymers, such as poly(lactic acid) (PLA), have gained interest as a substitute for conventional fossil-based polymers in biomedical and commodity applications. Its main features are its biodegradable nature, the decrease in the CO_2_ footprint associated with the product, and the non-toxic residues released during processing [3,4,5,6].

Despite its great potential, PLA still has limitations, such as its brittleness, its reduced service temperature range, and its high instability during processing where good melt strength is required. There is a large amount of research dedicated to solving these drawbacks with the aim of expanding its application window to become a commodity or even an engineering thermoplastic [7,8,9,10,11].

Blending PLA with other engineering soft polymers represents an industrially relevant strategy for developing bio-based formulations with tailored performances [12]. Specifically, numerous works report the investigation of melt blending PLA with polyamides (PA) [13,14,15,16,17,18]. However, the inherent immiscibility of this binary polymer system results in rather poor mechanical properties. To overcome the aforementioned issue, Pai et al. [14] and Patel et al. [18] reported the first attempts to compatibilize PLA/PA blends by adding titanium isopropoxide and a low molecular weight epoxy resin, respectively. Unfortunately, a high PA content (≥50%) is needed to change the blend morphology from a droplet-matrix to co-continuous in order to improve the mechanical properties [15,17,18]. Therefore, PLA/PA blends with a predominant PLA content still exhibit brittleness. Polymer blends exhibiting a fine-tuned morphology with a significantly reduced droplet size of the minor phase are promising due to their enhanced toughness in comparison to coarse sea-island morphologies. Indeed, the challenge is not only the compatibility of phases but also the control of the resulting morphology after processing, which contributes to determine the mechanical performance. Initially, a refined droplet morphology of the minor phase should be better, in terms of enhanced toughness in comparison to a coarse sea-island morphology. However, it has been demonstrated that the generation during processing of a suitably compatibilized and oriented fibril morphology of the dispersed phase could generate a mechanical reinforcing effect [19].

Among all available strategies that enable a processing-controlled morphology, the viscosity ratio of the parent polymers in blends is that which is considered in industrial practice. Indeed, using PLA as a matrix with enhanced melt viscosity and melt elasticity through reactive extrusion (using a styrene-acrylic multi-functional epoxide reactive agent) promotes a more homogeneous PA microstructure with improved interfacial adhesion, thus promoting a nucleant effect in the PLA phase [20]. Based on the work of Walha et al. [21], this enhanced feature could be attributed to the reaction between the unreacted epoxy groups present in modified PLA and the amine chain ends of PA.

Along with many types of aliphatic polyesters, PLA is subject to some thermal decomposition above its melting temperature, especially during processing. Radical and non-radical reactions have been proposed to explain the various complex mechanisms that could occur during processing that lead to a reduction in the molecular weight and viscosity. As a result, a general decrease in the material properties is expected. Yu et al. (2003) [22] argued that thermal and hydrolysis reactions for biocopolymers could be generated by random chain scission reactions of the ester groups. Coupled with this mechanism, intra- and inter-molecular transesterification reactions could also cause a drop in molecular weight at longer reaction times.

The ever-increasing commercial importance of polymeric materials has aroused continuous interest in their thermal stability. The kinetic modeling of the decomposition process plays a central role in many of these studies, as it is crucial for an accurate prediction of the material behavior under different working conditions [23,24,25,26,27,28,29,30,31]. A precise prediction requires knowledge of the so-called kinetic triplet: the activation energy, pre-exponential factor, and kinetic model. The latter parameter, also known as conversion function, *f*(*α*), is an algebraic expression that is associated with the mathematical model that describes the kinetics of solid-state reactions. Therefore, the kinetic analysis also provides some understanding of the mechanism of the reaction under study. Knowledge of the kinetic model and the mechanism of thermal degradation of macromolecules is very helpful in the study of the thermal stability of polymers [32,33,34].

The goal of this paper was to determine the thermal stability of PLA/PA bioblends containing between 50% and 90% rheologically modified PLA. Moreover, empirical (n-order and autocatalytic) and theoretical (R1, F1, D1, R2, F2, D2, R3, F3, D3, and random scission) kinetic models were tested in order to elucidate the best mechanism to describe the thermal degradation of PLA within these bioblends. The variation in activation energy values with PA content was also assessed.

## 2. Theoretical Background

Thermally stimulated solid-state reactions, such as thermal decompositions, are heterogeneous processes. The reaction rates of such processes can be kinetically described when they take place under conditions that are far from equilibrium, by the following expression:(1)$$\frac{d\alpha }{dt}=k\;f\left(\alpha \right)=A\;exp\left(-\frac{E}{RT}\right)f\left(\alpha \right)$$
where *dα*/*dt* is the reaction rate, *k* is the kinetic constant, A is the Arrhenius frequency factor, *R* is the gas constant, *E* is the activation energy, *α* is the reacted fraction or conversion, *T* is the process temperature, and *f*(*α*) accounts for the reaction rate dependence on *α*. The conversion function, *f*(*α*), describes the dependence of the reaction rate with the process mechanism.

When integrating Equation (1) and truncating the infinite series to the second term, the linear integral equation is derived for experiments carried out at a constant heating rate (β=dT/dt). This equation is called the general analytical equation (GAE), developed by Carrasco [35].
(2)$$ln\left[\beta \;\frac{g\left(\alpha \right)}{T^{2}\left(1-\frac{2RT}{E}\right)}\right]=ln\frac{AR}{E}-\frac{E}{R}\;\frac{1}{T}$$
where
(3)$$g\left(\alpha \right)=\int_{0}^{\alpha }\frac{d\alpha }{f\left(\alpha \right)}$$

It must be considered that the activation energy calculation, through Equation (2), requires an iterative procedure given that *E* is needed to evaluate the first member of the equation. The *E* value determined from the slope was introduced in the first member. The new *E* value obtained from the slope was again introduced in the first member and so on, until reaching a constant *E* value.

Different conversion functions are reported in literature for describing the kinetic mechanism of solid-state reactions. These mechanisms are proposed to consider different geometrical assumptions for the shape of the material particles (spherical, cylindrical, and planar) and driving forces (interface growth, diffusion, nucleation, and growth of nuclei). Table 1 shows the equations used for the linear regression analysis of the most common solid-state theoretical mechanisms (R1, R2, R3, F1, F2, F3, D1, D2, D3, and random scission). The random scission mechanism is applied to L = 2, where L is the minimum length of the polymer that is not volatile. For L ≥ 3, there is no symbolic solution and an iterative procedure is required. The conversion functions, *f*(*α*), assume idealized models, which may not be necessarily fulfilled in real systems. On the other hand, empirical kinetic models are also proposed: n-order and autocatalytic. The exponents *n* = 0.550 (for n-order kinetics), and *n* = 0.771 and *m* = 0.244 (for autocatalytic kinetics) were previously optimized for the thermal degradation of PLA [36]. The activation energy and frequency factor values for each kinetic model were calculated using Equations (2) and (3).

## 3. Materials and Methods

### 3.1. Materials

A commercially extrusion-grade PLA (Ingeo 4032DR; D lactide content: 2%) was purchased from Natureworks (Arendonk, Belgium) and used as received.

SAmfE reactive agent, namely Joncryl ADR-4300FR, was kindly supplied by BASF (Ludwigshafen, Germany) with an epoxy equivalent weight of 433 Da and a functionality of about 12.

The predominantly bio-based PA10.10 was produced by Dupont (Midland, MI, USA) under the trade name Zytel RS LC1000 BK385.

### 3.2. Reactive Extrusion and Bioblend Preparation

Bioblends were prepared by using a two-step process. Prior to processing, raw PLA was dried at 80 °C for 4 h in a Piovan (DSN506HE, Venice, Italy) hopper-dryer (dew point = −40 °C). SAmfE flakes were vacuum-dried overnight at RT over silica gel. The enhancement of the PLA melt properties was achieved through reactive extrusion using a corotating twin-screw extruder with a screw diameter of 25 mm (L/D = 36) (KNETER 25 × 24D, Collin GmbH, Ebersberg, Germany) and a nominal SAmfE weight content of 0.6%. The seven heating zones were set to 45, 165, 165, 170, 180, 190, 190 °C from feeding to die zones.

The screw speed was set to 35 rpm, leading to a residence time of 4.1 min. The extrudate was water-cooled and pelletized. Then, five PLA/PA blends, covering the 10–50% PA weight composition range, were prepared by melt mixing, using a Brabender batch mixer (Brabender Plastic-Corder W50EHT, Brabender GmbH & Co., Duisburg, Germany) operated at 210 °C and 50 rpm for 12 min. The obtained materials were further compression molded into 0.6 mm thick plates in an IQAP LAP PL-15 hot plate press (IQAP Masterbatch S.L., Barcelona, Spain) for 5 min at 210 °C and 4 MPa and then cooled at −50 °C/min. Prior to processing, PLA and PA were vacuum dried overnight at 80 °C over silica gel.

### 3.3. Thermal Characterization

TGA data were processed on a Mettler Toledo thermogravimetric analyzer, model TGA-SDTA851. Samples of 20 mg were heated at various linear heating rates (2.5, 5, and 10 K/min) from room temperature to 600 °C, under a dry nitrogen gas flow rate of 40 cm^3^/min. Two replicates were scanned and errors were lower than 1.5%.

## 4. Results and Discussion

Figure 1 shows the experimental curves recorded for the thermal degradation of PLA (produced by reactive extrusion), PA, and two PLA/PA bioblends (with PLA weight contents of 50% and 90%), under a nominal linear heating rate of 10 K/min. It is clear that PA is significantly more resistant to thermal degradation compared to PLA. TG curves of PLA and PA are sigmoidal and they present a unique profile of decomposition. Therefore, the thermal degradation of these pure polymers presents a unique value of activation energy within the entire range of conversion. Contrarily, the TG curves of PLA/PA bioblends clearly exhibit three different zones of decomposition: a first step, where PLA degrades, a second step, where the decomposition of PLA and PA takes place simultaneously, and a third phase, where residual PA degrades until the total disappearance of the organic material. When comparing the first stages of the TG curves of the 90/10 and 50/50 bioblends, it can be concluded that the bioblend with a higher PA content is a more thermally resistant material. Therefore, the presence of PA in higher proportions clearly protects the matrix of PLA. From the TG data, it was possible to evaluate the conversion and the conversion derivative by taking into account the inorganic residual material at 600 °C, which was lower than 2%.

Figure 2 shows the conversion derivative values of PLA and PA as a function of temperature. This graphic clearly reveals that PLA and PA degrade in a completely different temperature range: 290–380 °C for PLA and 380–490 °C for PA. This finding is very important because it means that we were able to study the thermal degradation of PLA without any decomposition of PA. For this reason, it was possible to study the influence of PA on the thermal degradation of the PLA matrix. Given that the different materials contain 10–50% PA, the kinetic study of the degradation of PLA was considered for conversions of PLA lower than 0.4. Within this interval (α = 0–0.4), no degradation of PA can occur, thus allowing for the specific study of the thermal stability of PLA in the presence of various proportions of PA.

Figure 3 shows the experimental conversion curve for the bioblend containing 50/50 PLA/PA (under a nominal linear heating rate of 10 K/min) and the theoretical conversion curve constructed by a linear combination of PLA and PA experimental curves. The difference between these curves was caused by the additional melting process needed to prepare the bioblend and the development of a new tridimensional structure, as a consequence of interactions between the molecular chains of PLA and PA. At α < 0.25, the bioblend is less resistant to heat (due to the melting process), whereas at α > 0.25, it is more resistant (due to interactions between PLA and PA). If we consider the degradation of single components equivalent to 5% of thermal degradation (in order to “take into account the bioblending process”), the “modified theoretical” curve clearly shows that thermal degradation is higher for the sum of single components compared to that of the bioblend, thus demonstrating the protective effect of PA on the PLA matrix.

From Figure 1, it was possible to obtain various decomposition temperatures, i.e., the onset decomposition (*T*_5_) and final decomposition (*T*_95_) temperatures, defined as the temperatures at which 5% and 95% of the mass was lost, respectively, as shown in Table 2. The values of *T*_5_ for the bioblends increased with PA content, from 315.8 °C to 321.2 °C when the PA content increased from 10% to 50%. On the other hand, the values of *T*_95_ ranged from 455 °C to 554 °C, with an erratic variation as a function of the PA content. Table 2 also shows other thermal parameters, i.e., conversion (*α_m_*), conversion derivative ((*dα*/*dT*)*_m_*), and temperature (*T_m_*) at the maximum decomposition rate of PLA.

It is clear that the presence of PA considerably modified the conversion values at the maximum rate from 45% to 27% when the PA content increased from 10% to 50%. This shift was essentially due to the variable content of PLA in the bioblends and also to the presence of interactions between the PLA matrix and PA.

Table 2 also contains the temperature (*T_r_*) and conversion derivative (*dα*/*dT*)*_r_*) values at *α* = *α_r_* (reference conversion). These values are necessary to construct standardized conversion functions, as is shown later. Usually, reference conversion is taken as being equal to 0.5. This is correct for pure polymers. In the case of mixed polymers, this value must not be considered. For example, for the 50/50 bioblend, a conversion of 0.5 means the total decomposition of PLA and this cannot be considered as a reference of the PLA thermal decomposition. For this reason, a new reference conversion was defined in this work: *α_r_* = 0.5 x, where x is the mass fraction of PLA in bioblends. Therefore, the reference conversions are 0.45, 0.40, 0.35, 0.30, and 0.25 for bioblends containing 90%, 80%, 70%, 60%, and 50% PLA. Curiously, these reference conversion values are very near to the conversion values at the maximum decomposition rate.

Figure 4 shows the results of the general analytical equation (GAE) for three theoretical conventional mechanisms (F2, R2, and D2) for the thermal degradation of the bioblend containing 70% PLA, under a nominal linear heating rate of 10 K/min. The experimental data fit was excellent (*r*^2^ > 0.99) for the three kinetic models. The activation energy values evaluated are shown in Table 3 (mean value for the three heating rates ± confidence interval). The activation energy values for the thermal degradation of PLA depend on the considered mechanism and the content of PA. These E values ranged, for the various bioblends, as a function of the reaction mechanism as follows: 236–311 kJ/mol (n-order), 192–254 kJ/mol (autocatalytic), 154–205 kJ/mol (random scission), 253–334 kJ/mol (F1), 216–284 kJ/mol (R1), 441–578 kJ/mol (D1), 294–389 kJ/mol (F2), 234–308 kJ/mol (R2), 464–610 kJ/mol (D2), 340–450 kJ/mol (F3), 240–317 kJ/mol (R3), and 490–643 kJ/mol (D3). Therefore, the activation energy values are highly influenced by the presence of PA for each mechanism. Diffusion (D1, D2, and D3) and the F2 mechanisms had the highest activation energies, whereas random scission, autocatalytic, and R1 had the lowest activation energies.

There was a compensation factor between the frequency factor and activation energy for all the empirical and theoretical kinetic models and for all the materials under study. Figure 5 shows an excellent linear relationship between LnA and E (for TG experiments carried out at a nominal heating rate of 10 K/min). The equation relating these two kinetic parameters is LnA (s^−1^) = −4.15 + 0.19 E (kJ/mol). This means that an increase in the activation energy (i.e., a higher energy barrier for the thermal degradation) leads to an increase in the frequency factor (i.e., a higher probability to be decomposed). This compensation factor was also observed for the other heating rates used in this work.

When summarizing the results of the linear regressions, all the mechanisms led to regression coefficients higher than 0.99 for all the bioblends. This means that all the mechanisms are satisfactory, from a mathematical point-of view, to represent the kinetics of the thermal degradation of the samples considered in this study. Therefore, it was necessary to employ a method that is able to discern the best mechanism (and then it became possible to evaluate the activation energy responsible for splitting the macromolecules, which occurred during the thermal degradation). For this reason, an isoconversional kinetic analysis was carried out.

Following the ICTAC recommendations [37], the Kissinger–Akahira–Sunose method was employed:(4)$$ln\left[\;\frac{\beta_{i}}{T_{\alpha ,i}^{2}}\right]=Const-\frac{E_{\alpha }}{R}\;\frac{1}{T_{\alpha }}$$

From this method, it was possible to evaluate the isoconversional values of the activation energy without assuming any particular form of reaction model. For this reason, isoconversional methods are frequently called “model-free” methods. The nominal heating rates used for this analysis were 2.5, 5, and 10 K/min. As reported by Vyazovkin et al. [37], determination of the reaction model requires that the variation in *E_α_* with *α* to be negligible so that *E_α_* dependence can be replaced with a single average value, *E_o_*. In our study, no sample exhibited a significant dependence of activation energy with conversion.

In order to elucidate the best reaction mechanism, standardized conversion functions were successfully used to compare the experimental and theoretical data relative to the reaction mechanisms. Theoretical values were determined using conversion functions, as shown in Table 1, and they depended only on the reaction mechanism. On the other hand, experimental values were calculated by means of the following equation:(5)$$\frac{f\left(\alpha \right)}{f\left(\alpha_{r}\right)}=\frac{\left(d\alpha /dT\right)}{{\left(d\alpha /dT\right)}_{r}}\;exp\left[\frac{E}{R}\;\left(\frac{1}{T}-\frac{1}{T_{r}}\right)\right]$$
where $f\left(\alpha_{r}\right)$, ${\left(d\alpha /dT\right)}_{r}$, and $T_{r}$ are the conversion function, conversion derivative, and temperature at α = $\alpha_{r}$ (reference conversion), respectively. These reference conversions are reported in Table 2. The conversion derivative and temperature at  α = $\alpha_{r}$ are experimental values and *E* is the activation energy previously evaluated for each of the mechanisms considered, as shown in Table 3. Therefore, the experimental values depend on the activation energy values.

Figure 6 shows the variation in the standardized conversion functions with the conversion for two mechanisms: random scission and D1. The fitting between the experimental and theoretical values was excellent for the random scission mechanism, but this fitting was very poor when mechanism D1 was considered. These standardized conversion functions give valuable information about the most suitable mechanism. However, this information is only qualitative. In order to elucidate the most appropriate kinetic model, it was necessary to develop a quantitative procedure. For this reason, an index, the integral mean error (*IME*), was proposed in a previous work [38]. This index takes into consideration the mean area under the curve of absolute error of the standardized conversion function vs. conversion, and is defined as follows:(6)$$IME=\frac{\int_{0}^{\alpha }\left|\Delta_{f\left(\alpha \right)/f\left(\alpha_{r}\right)}\right|d\alpha }{\int_{0}^{\alpha }d\alpha }\cdot 100$$

Table 4 contains integral mean error (*IME*) values for all the mechanisms studied in this work. For each bioblend, the minimum *IME* values were found for the random scission mechanism (5.5–6.7%), which correspond with the lowest errors.

The activation energy values of the thermal decomposition of PLA, through a random scission mechanism, are those shown in Figure 7. These results indicate that the activation energy increases when the PA content increases because of the protecting effect of PA on the PLA matrix (especially for a PA content up to 30%). Indeed, the activation energy increased by 16 kJ/mol when the PA content increased from 10% to 20%, and it increased by 34 kJ/mol when the PA content increased from 20% to 30%. The activation energy remained almost constant when the PA content increased from 30% to 40%. Contrarily, an activation energy decrease of 25 kJ/mol was observed when the PA content increased from 40% to 50%; the latter was likely due to an inversion of phases caused by the high PA content.

In a previous work dealing with the morphology of PLA/PA bioblends, Cailloux et al. [39] stated that both droplet and elongated PA domains coexisted when the PA content was 30%. The morphology transition to co-continuous was completed when the PA content was further increased to 40%. Therefore, the bioblend morphology clearly has a protective effect against the thermal decomposition of PLA within the bioblend. Moreover, a decrease in the stiffness and strength and an increase in the strain were reported with an increasing PA content. On the other hand, diffusion mechanisms led to the highest *IME* values (27–88%), which are unacceptable. This is later corroborated by means of standardized conversion function plots and *y*(*α*) master plots.

In order to check the validity of the activation energy values obtained for the thermal degradation of PLA, *y*(*α*) master plots, as proposed by Criado et al. [40], were used:(7)$$y{\left(\alpha \right)}_{exptal}={\left(\frac{T}{T_{r}}\right)}^{2}\;\frac{\left(d\alpha /dT\right)}{{\left(d\alpha /dT\right)}_{r}}$$
(8)$$y{\left(\alpha \right)}_{theor}=\frac{f\left(\alpha \right)\;g\left(\alpha \right)}{f\left(\alpha_{r}\right)\;g\left(\alpha_{r}\right)}$$

These master plots are very interesting because experimental data are not dependent on activation energy values. Figure 8 shows *y*(*α*) master plots for the thermal degradation of PLA for the bioblend containing 90% PLA. These findings corroborate the previously reported results, i.e., the best reaction mechanism for the thermal degradation of PLA is random scission, whereas D1 and F2 mechanisms showed poor fitting between experimental and theoretical data. The rest of the mechanisms also provided poor fitting. In conclusion, the best mechanism is that of random scission of macromolecular chains, as confirmed through three different pieces of evidence: the standardized conversion functions, *IME* values, and *y*(*α*) master plots.

The variation in conversion with temperature for the thermal degradation of PLA, through a random scission mechanism, is evaluated as follows:(9)$$\alpha ={\left[1-exp\left[-\frac{A\;R\;T^{2}}{\beta \;E}\;\left(1-\frac{2RT}{E}\right)\;exp\left(-\frac{E}{RT}\right)\right]\right]}^{2}$$

Figure 9 shows the variation in conversion with temperature when the random scission mechanism was considered for the bioblend containing 90% PLA. The fitting between the experimental and theoretical data was excellent, thus confirming that random scission is the best mechanism with which to explain the thermal degradation of PLA in PLA/PA bioblends.

## 5. Conclusions

The thermal stability and potential degradation mechanisms of PLA in PLA/bio-based PA 10.10 were analyzed. In this case, a matrix of a rheologically modified PLA from reactive extrusion was employed, with the addition of 10–50% of biobased PA as a second phase. PLA/PA bioblends (with a predominantly biosourced PA10.10) containing 10–50% PA were prepared by melt blending in order to overcome the extreme brittleness of PLA.

The temperature at which 5% of the mass was lost (*T*_5_) increased from 315 to 321 °C when the PA content increased from 10% to 50% (at a nominal heating rate of 10 K/min). On the other hand, the temperature at which 95% of the mass was lost (*T*_95_) varied erratically in the range 455–540 °C. Similar conclusions were obtained at the other heating rates employed in this study.

The general analytical equation (GAE) was used in order to evaluate the kinetic parameters of the thermal degradation of PLA (at conversions lower than 0.4). Various empirical and theoretical solid-state mechanisms were tested to elucidate the best kinetic model: n-order, autocatalytic, random scission, F1, F2, F3, R1, R2, R3, D1, D2, and D3. Three different methodologies were used for this: the standardized conversion functions, *IME* (integral mean error) indexes, and *y*(*α*) master plots, which revealed that the most probable mechanism for the thermal degradation of PLA was random scission of the macromolecular chains. The activation energies obtained were 154, 170, 204, 205, and 180 kJ/mol for the bioblends containing 10%, 20%, 30%, 40%, and 50% PA. These results indicate that the activation energy increases when the PA content increases because of the protecting effect of PA on the PLA matrix (especially for PA contents up to 30%). Indeed, the activation energy increased by 16 kJ/mol when the PA content increased from 10% to 20%, and it increased by 34 kJ/mol when the PA content increased from 20% to 30%. The activation energy remained almost constant when the PA content increased from 30% to 40%. Contrarily, an activation energy decrease of 25 kJ/mol was observed when the PA content increased from 40% to 50%; the latter was likely due to an inversion of phases caused by the high PA content. Therefore, the bioblend containing 30% PLA exhibited an excellent thermal resistance against degradation. This result is in accordance with previous rheological and morphological analyses.

## Figures and Tables

**Figure 1 polymers-13-03996-f001:**
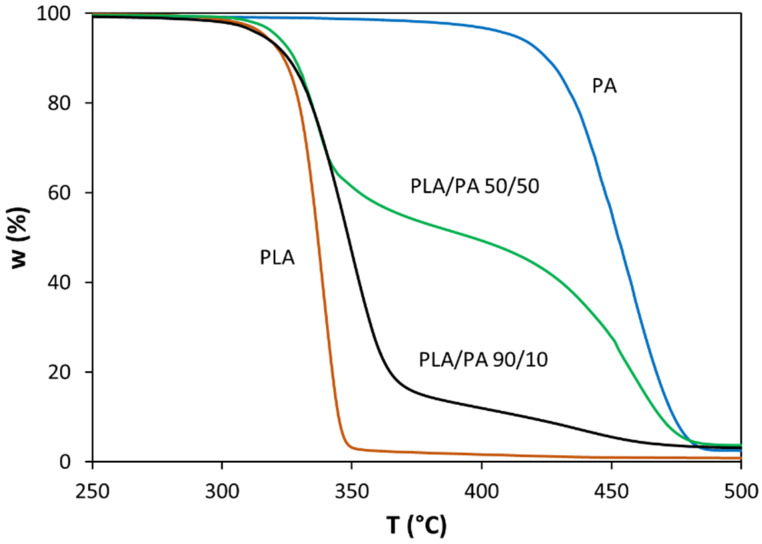
TG curves (i.e., variation in residual weight as a function of temperature) for PLA, PA, and bioblends containing 50% and 90% PLA (nominal heating rate = 10 K/min).

**Figure 2 polymers-13-03996-f002:**
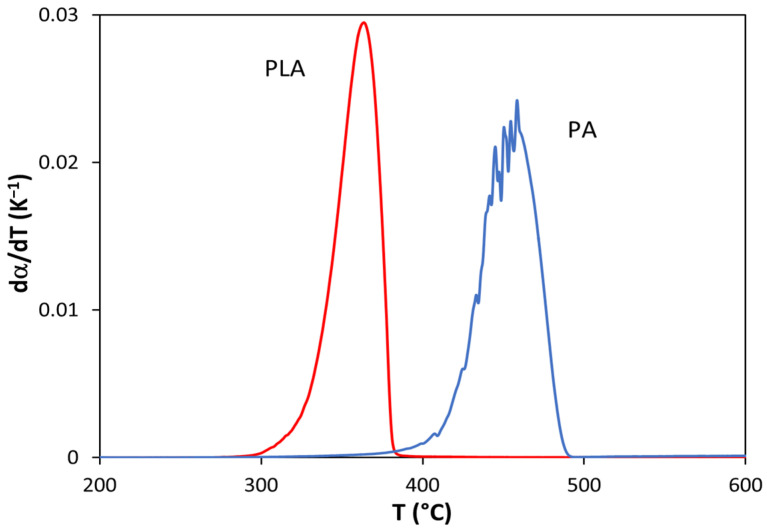
Variation in the conversion derivative of PLA and PA as a function of temperature (nominal heating rate = 10 K/min).

**Figure 3 polymers-13-03996-f003:**
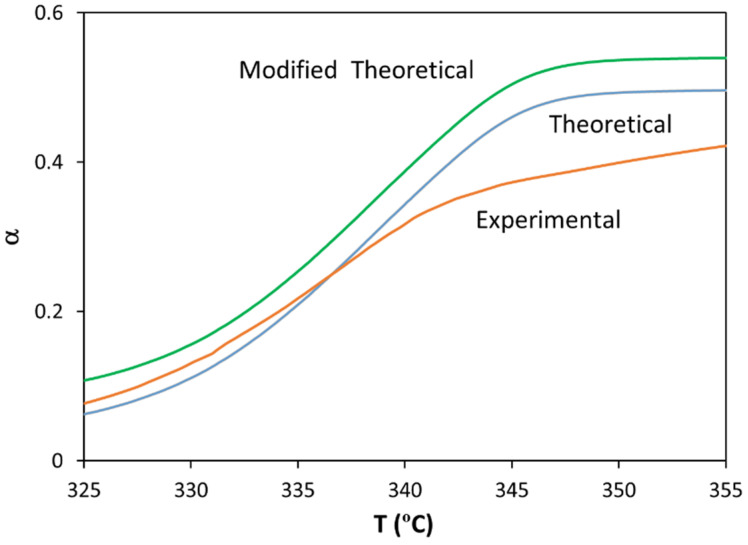
Variation in conversion with temperature for the bioblend containing 50% PLA (nominal heating rate = 10 K/min).

**Figure 4 polymers-13-03996-f004:**
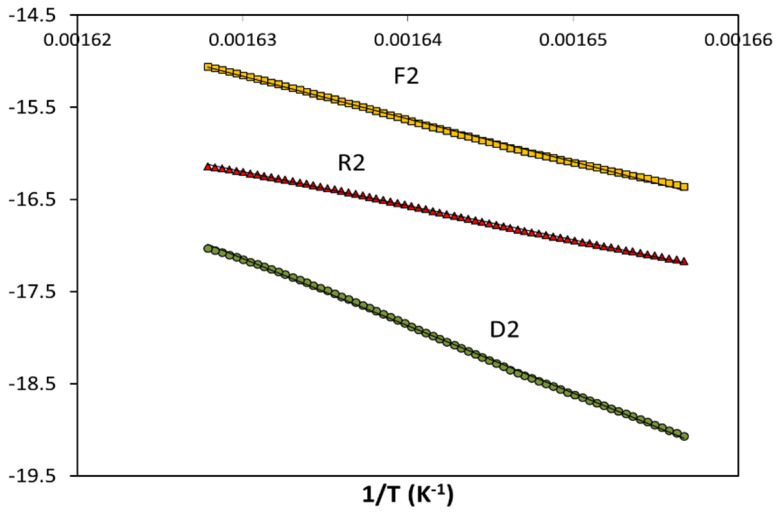
Integral kinetic analysis by means of the general analytical equation (GAE) for the thermal degradation of the bioblend containing 70% PLA using three theoretical conventional mechanisms (F2, R2, and D2) (nominal heating rate = 10 K/min).

**Figure 5 polymers-13-03996-f005:**
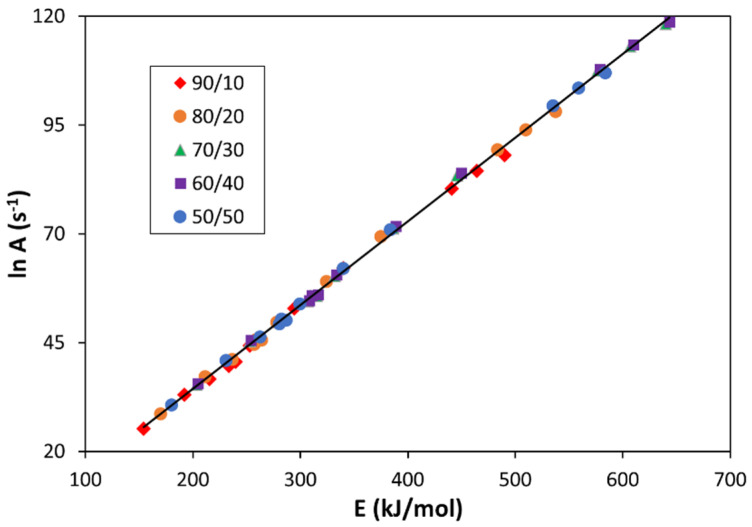
Compensation factor between the frequency factor and activation energy for all the empirical and theoretical kinetic models studied and for all the bioblends analyzed (nominal heating rate = 10 K/min).

**Figure 6 polymers-13-03996-f006:**
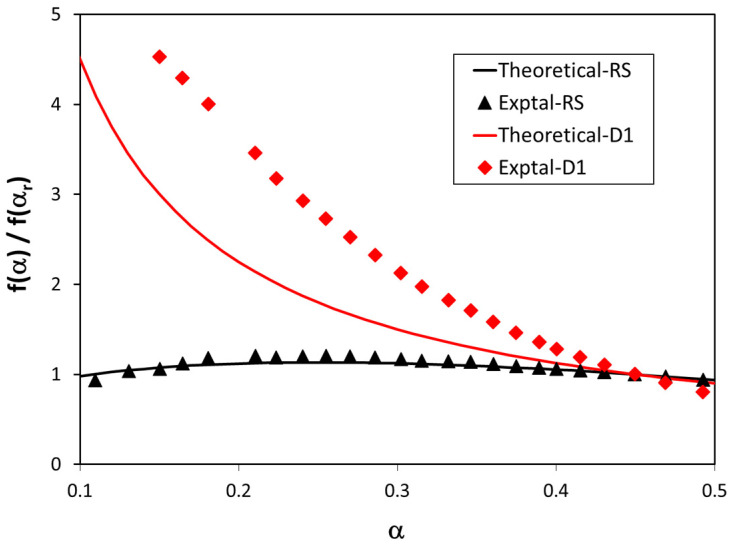
Variation in standardized conversion functions as a function of conversion for the thermal degradation of PLA in the bioblend containing 90% PLA, when considering two reaction mechanisms: random scission and D1.

**Figure 7 polymers-13-03996-f007:**
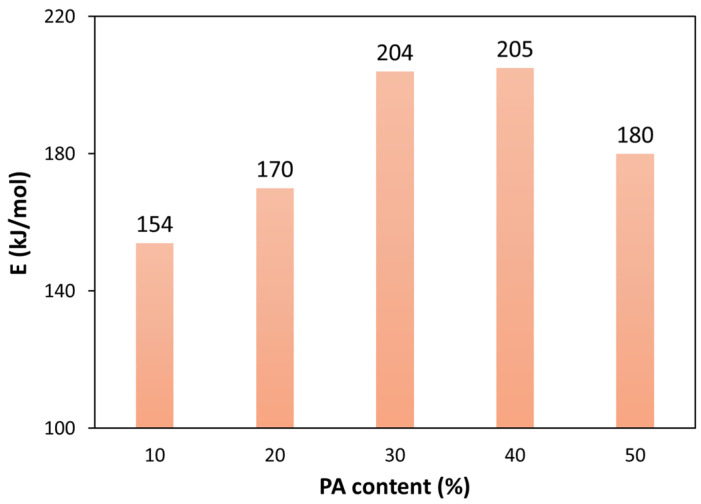
Activation energy values of the thermal decomposition of PLA through a random scission mechanism: the influence of PA content.

**Figure 8 polymers-13-03996-f008:**
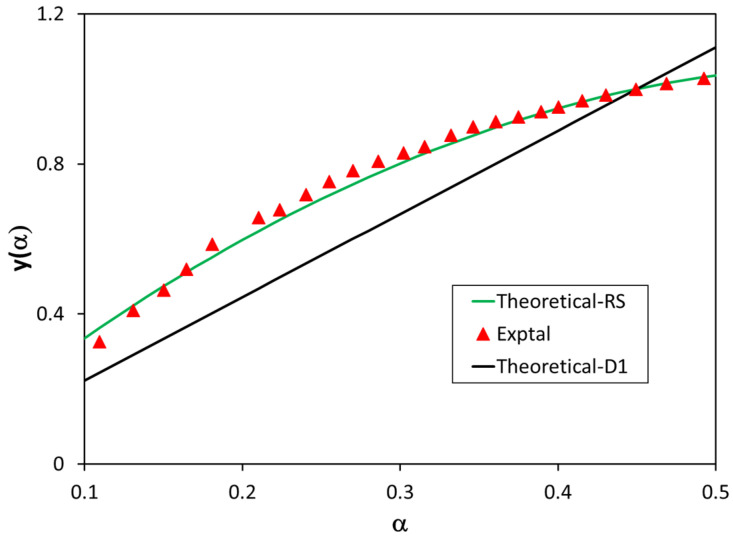
Comparison between experimental and theoretical values of *y*(*α*) master plots for random scission (RS) and D1 mechanisms for the bioblend containing 90% PLA.

**Figure 9 polymers-13-03996-f009:**
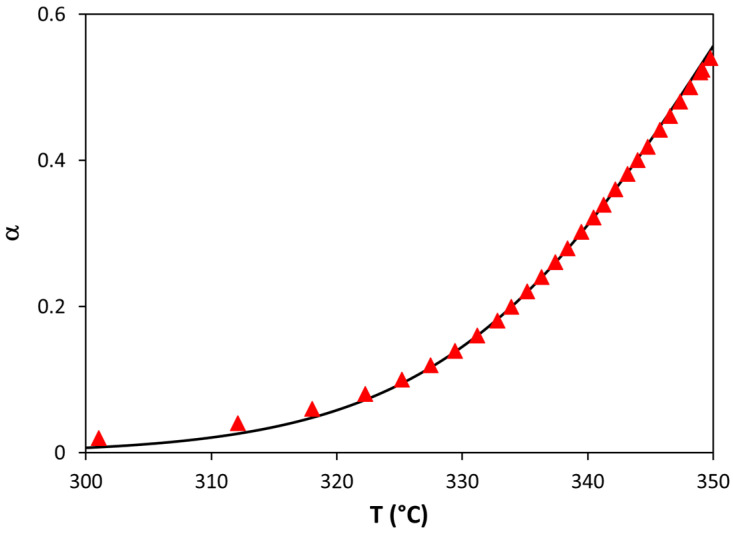
Comparison between experimental and random scission theoretical values of conversion for the thermal degradation of PLA in the bioblend containing 90% PLA.

**Table 1 polymers-13-03996-t001:** Integral kinetic equations for various solid-state mechanisms.

Mechanism	f(α)	General Analytical Equation in Linear Form
n-order	${\left(1-\alpha \right)}^{n}$	$ln\left[\beta \;\frac{1-{\left(1-\alpha \right)}^{1-n}}{\left(1-n\right)\;T^{2}\left(1-\frac{2RT}{E}\right)}\right]=ln\frac{AR}{E}-\frac{E}{R}\frac{1}{\;T}$
Autocatalytic	${\left(1-\alpha \right)}^{n}\alpha^{m}$	$ln\left[\beta \;\frac{\frac{1-{\left(1-\alpha \right)}^{1-n}}{1-n}+\frac{\alpha^{1-m}}{1-m}\;}{T^{2}\left(1-\frac{2RT}{E}\right)}\right]=ln\frac{AR}{E}-\frac{E}{R}\;\frac{1}{\;T}$
Random scission	$2\;\left(\alpha^{1/2}-\alpha \right)$	$ln\left[\beta \;\frac{-ln\left(1-\alpha^{1/2}\right)}{T^{2}\left(1-\frac{2RT}{E}\right)}\right]=ln\frac{AR}{E}-\frac{E}{R}\frac{1}{\;T}$
R1	1	$ln\left[\beta \;\frac{\alpha }{\;T^{2}\left(1-\frac{2RT}{E}\right)}\right]=ln\frac{AR}{E}-\frac{E}{R}\frac{1}{\;T}$
R2	$2\;{\left(1-\alpha \right)}^{1/2}$	$ln\left[\beta \;\frac{1-{\left(1-\alpha \right)}^{1/2}\;}{T^{2}\left(1-\frac{2RT}{E}\right)}\right]=ln\frac{AR}{E}-\frac{E}{R}\;\frac{1}{\;T}$
R3	$3\;{\left(1-\alpha \right)}^{2/3}$	$ln\left[\beta \;\frac{1-{\left(1-\alpha \right)}^{1/3}}{T^{2}\left(1-\frac{2RT}{E}\right)}\right]=ln\frac{AR}{E}-\frac{E}{R}\frac{1}{\;T}$
F1	(1−α)	$ln\left[\beta \;\frac{-ln\left(1-\alpha \right)}{T^{2}\left(1-\frac{2RT}{E}\right)}\right]=ln\frac{AR}{E}-\frac{E}{R}\frac{1}{\;T}$
F2	${\left(1-\alpha \right)}^{2}$	$ln\left[\beta \;\frac{\left(\frac{\alpha }{1-\alpha }\right)}{T^{2}\left(1-\frac{2RT}{E}\right)}\right]=ln\frac{AR}{E}-\frac{E}{R}\frac{1}{\;T}$
F3	${\left(1-\alpha \right)}^{3}$	$ln\left[\beta \;\frac{{\left(1-\alpha \right)}^{-2}-1}{2\;T^{2}\left(1-\frac{2RT}{E}\right)}\right]=ln\frac{AR}{E}-\frac{E}{R}\frac{1}{\;T}$
D1	$\frac{1}{2\alpha }$	$ln\left[\beta \;\frac{\alpha^{2}}{T^{2}\left(1-\frac{2RT}{E}\right)}\right]=ln\frac{AR}{E}-\frac{E}{R}\frac{1}{\;T}$
D2	$\frac{1}{-ln\left(1-\alpha \right)}$	$ln\left[\beta \;\frac{\left(1-\alpha \right)\;ln\left(1-\alpha \right)+\alpha }{T^{2}\left(1-\frac{2RT}{E}\right)}\right]=ln\frac{AR}{E}-\frac{E}{R}\frac{1}{\;T}$
D3	$\frac{3\;{\left(1-\alpha \right)}^{2/3}\;}{2\;\left[1-{\left(1-\alpha \right)}^{1/3}\right]}$	$ln\left[\beta \;\frac{1+{\left(1-\alpha \right)}^{2/3}-2\;{\left(1-\alpha \right)}^{1/3}}{T^{2}\left(1-\frac{2RT}{E}\right)}\right]=ln\frac{AR}{E}-\frac{E}{R}\frac{1}{\;T}$

**Table 2 polymers-13-03996-t002:** Decomposition temperatures of the PLA/PA bioblends, at a nominal heating rate of 10 K/min.

	*T*_5_ (°C)	*T*_95_ (°C)	*α_m_*	*T_m_* (°C)	(*dα*/*dT*)*_m_* (K^−1^)	*α_r_*	*T_r_* (°C)	(*dα*/*dT*)*_r_* (K^−1^)
PLA/PA 90/10	315.8	455.0	0.54	349.8	2.39·10^−2^	0.45	346.1	2.38·10^−2^
PLA/PA 80/20	317.4	554.4	0.43	343.1	2.49·10^−2^	0.40	341.8	2.46·10^−2^
PLA/PA 70/30	318.8	540.2	0.35	339.2	2.83·10^−2^	0.35	339.3	1.76·10^−2^
PLA/PA 60/40	320.5	484.1	0.30	338.5	2.69·10^−2^	0.30	338.4	0.49·10^−2^
PLA/PA 50/50	321.2	479.2	0.27	337.6	2.01·10^−2^	0.25	336.6	0.18·10^−2^

**Table 3 polymers-13-03996-t003:** Mean activation energies of the thermal degradation of PLA, obtained using the general analytical equation at various heating rates, for PLA/PA bioblends containing from 50% to 90% PLA.

	E (kJ/mol)				
Mechanism	90/10	80/20	70/30	60/40	50/50
n-order	236 ± 4	259 ± 3	309 ± 3	311 ± 3	283 ± 3
Autocatalytic	192 ± 3	212 ± 3	253 ± 3	254 ± 3	231 ± 2
Random scission	154 ± 2	170 ± 2	204 ± 3	205 ± 3	180 ± 2
F1	253 ± 4	278 ± 4	332 ± 4	334 ± 5	300 ± 4
R1	216 ± 4	237 ± 3	283 ± 3	284 ± 3	263 ± 3
D1	441 ± 5	484 ± 5	576 ± 6	578 ± 6	536 ± 6
F2	294 ± 4	324 ± 5	386 ± 5	389 ± 5	340 ± 4
R2	234 ± 4	257 ± 3	307 ± 3	308 ± 4	281 ± 3
D2	464 ± 6	510 ± 6	607 ± 6	610 ± 5	559 ± 6
F3	340 ± 5	375 ± 5	446 ± 6	450 ± 6	384 ± 5
R3	240 ± 3	264 ± 3	315 ± 4	317 ± 4	287 ± 3
D3	490 ± 7	538 ± 7	640 ± 8	643 ± 8	584 ± 7

**Table 4 polymers-13-03996-t004:** Integral mean error (*IME*) between experimental and theoretical data of *f*(*α*)/*f*(*α_r_*) for various mechanisms for the thermal degradation of PLA for bioblends containing from 50% to 90% PLA.

	*IME* (%)				
Mechanism	90/10	80/20	70/30	60/40	50/50
n-order	28.7 ± 0.7	22.9 ± 0.6	7.9 ± 0.2	7.8 ± 0.2	21.1 ± 0.5
Autocatalytic	20.3 ± 0.6	17.7 ± 0.4	7.1 ± 0.2	7.0 ± 0.2	40.1 ± 0.5
Random scission	5.5 ± 0.2	6.0 ± 0.2	6.2 ± 0.2	6.2 ± 0.2	6.7 ± 0.3
F1	20.3 ± 0.5	18.1 ± 0.4	7.2 ± 0.2	7.6 ± 0.3	25.7 ± 0.4
R1	37.2 ± 0.8	28.1 ± 0.6	11.1 ± 0.3	9.4 ± 0.3	19.5 ± 0.4
D1	87.8 ± 0.9	57.5 ± 0.8	14.5 ± 0.3	17.3 ± 0.4	27.3 ± 0.5
F2	5.8 ± 0.3	6.8 ± 0.2	8.9 ± 0.3	9.0 ± 0.3	25.7 ± 0.5
R2	29.5 ± 0.6	23.4 ± 0.5	8.9 ± 0.3	7.8 ± 0.2	20.9 ± 0.4
D2	81.3 ± 0.9	54.0 ± 0.7	11.0 ± 0.3	17.1 ± 0.4	28.7 ± 0.6
F3	40.3 ± 0.7	12.2 ± 0.4	15.7 ± 0.4	12.3 ± 0.3	28.0 ± 0.7
R3	26.7 ± 0.5	21.7 ± 0.4	8.6 ± 0.3	8.0 ± 0.3	21.4 ± 0.6
D3	71.1 ± 0.7	48.9 ± 0.8	10.8 ± 0.3	17.0 ± 0.4	30.0 ± 0.7

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
