# Peer review of "Kinetics of the Thermal Degradation of Poly(lactic acid) and Polyamide Bioblends"

_polymers, 2021, doi:10.3390/polym13223996_

Round 1
Reviewer 1 Report
Kinetics of the thermal degradation of poly(lactic acid) and pol-yamide bioblends
Comments:
- Its better author should write a nomenclature, to provide the list of symbols, including Greek letters and abbreviation.
- Application must be more explained and make your work synchronize with the real-world applications.
- Author must comment(s) on the previous and explain why your work is different? Justify your work that why it should be published in polymers.
- How did you get Eq. (1)? What is the physical meaning of this equation?
- If possible, provide a comparison to justify your work.
- English language must be improved according to the standard of POLYMER journal.
Author Response
We are enclosing a file.

Reviewer 2 Report
- In the introduction section, the Authors should describe in more detail the research on the improvement of compatibility of PLA/PA blends.
- The abbreviation for Equation is Eq., not Eqn.
- Figure 1. Authors should provide full axis captions, not only symbols that are not defined in the text of the article.
- The authors use the notation "β" as heating rate, which is nowhere defined in the text.
- Line 218. Is: conversion ( m), should be: conversion (αm).
- Equations 6-8 have a different text editing than the rest of the equations.
- Why did the authors include figure 9 in the Conclusion section? It is not a good place for it. There is no possibility to discuss it at this place. In addition, the Conclusion Section should be better supported by research results.
- The English language of the article should be checked by a Native Speaker.
Author Response
We are enclosing a file.

Round 2
Reviewer 1 Report
accepted
Reviewer 2 Report
The article has been significantly improved. I recommend publishing this article.